# Room-temperature Tamm-plasmon exciton-polaritons with a WSe$_2$ monolayer

Nils Lundt[1], Sebastian Klembt[1], Evgeniia Cherotchenko[2], Simon Betzold[1], Oliver Iff[1], Anton V. Nalitov[2], Martin Klaas[1], Christof P. Dietrich[1], Alexey V. Kavokin[2,3], Sven Höfling[1,4] & Christian Schneider[1]

Solid-state cavity quantum electrodynamics is a rapidly advancing field, which explores the frontiers of light–matter coupling. Metal-based approaches are of particular interest in this field, as they carry the potential to squeeze optical modes to spaces significantly below the diffraction limit. Transition metal dichalcogenides are ideally suited as the active material in cavity quantum electrodynamics, as they interact strongly with light at the ultimate monolayer limit. Here, we implement a Tamm-plasmon-polariton structure and study the coupling to a monolayer of WSe$_2$, hosting highly stable excitons. Exciton-polariton formation at room temperature is manifested in the characteristic energy–momentum dispersion relation studied in photoluminescence, featuring an anti-crossing between the exciton and photon modes with a Rabi-splitting of 23.5 meV. Creating polaritonic quasiparticles in monolithic, compact architectures with atomic monolayers under ambient conditions is a crucial step towards the exploration of nonlinearities, macroscopic coherence and advanced spinor physics with novel, low-mass bosons.

[1] Technische Physik and Wilhelm-Conrad-Röntgen Research Center for Complex Material Systems, Universität Würzburg, Am Hubland, D-97074 Würzburg, Germany. [2] Physics and Astronomy School, University of Southampton, Highfield, Southampton SO171BJ, UK. [3] SPIN-CNR, Viale del Politecnico 1, I-00133 Rome, Italy. [4] SUPA, School of Physics and Astronomy, University of St Andrews, St Andrews KY 16 9SS, UK. Correspondence and requests for materials should be addressed to N.L. (email: nils.lundt@physik.uni-wuerzburg.de) or to C.S. (email: christian.schneider@physik.uni-wuerzburg.de).

With the first reports of atomic monolayer materials exfoliated from a graphite block, a genuine breakthrough in physics was triggered. However, from the point of view of opto-electronic applications, pristine graphene has its limitations, as it does not have a (direct) bandgap. Two-dimensional atomic crystals of transition metal dichalcogenides (TMDCs), compounds of a $MX_2$ stoichiometry (M being a transition metal, X a chalcogenide), seem to be much more promising[1–6], as monolayers of some TMDCs have a direct bandgap on the order of 1.6–2.1 eV[2,7]. Furthermore, the combination of large exciton-binding energies up to 550 meV[2,8], large oscillator strength, the possible absence of structural disorder and intriguing spinor and polarization properties[9–12] has recently placed sheets of TMDCs in the focus of solid-state cavity quantum electrodynamics and polaritonics. Polariton formation can be observed in the strong light–matter coupling regime, which becomes accessible in high quality, or ultra-compact photonic structures, such as dielectric microcavities or plasmonic architectures with embedded emitters comprising large oscillator strengths[13,14]. Once the light–matter coupling strength in such a system exceeds dissipation and dephasing, the hybridization of light and matter excitations leads to the formation of exciton-polaritons[13]. These composite quasi-particles have very appealing physical properties. Polaritons can travel over macroscopic distances at high speed (∼1% of the speed of light[15]) and, due to the inherited matter component, interactions between polaritons are notable. This puts them in the focus of nonlinear optics, collective bosonic phenomena and integrated photonics. Exciton-polaritons are bosons with a very low (and tailorable) effective mass and are therefore almost ideal candidates to study Bose–Einstein condensation phenomena at elevated temperatures. A serious drawback for this field, which circumvents a more efficient exploitation of polaritons, are the limited thermal stability of excitons in most of the III/V materials, strong disorder and defects in II–VI multilayer structures and exciton bleaching in organic polymers. TMDCs monolayers have the potential to overcome these drawbacks and are therefore considered as a highly promising material platform for light–matter interaction experiments[1].

Recently, the physics of strong light–matter coupling between a single flake of $MoS_2$ and a cavity resonance in a Fabry–Perot resonator structure was discussed[16]. However, the comparably broad photoluminescence (PL) emission of the monolayer used for these findings (60 meV[16]), which was grown by chemical vapour deposition, render the unambiguous identification of the full characteristic polariton dispersion relation, in particular in non-resonant PL experiments challenging. Unambiguous polariton formation with a single monolayer of $MoSe_2$ was subsequently demonstrated at cryogenic temperatures[17], enabled by its narrow linewidth (11 meV at 4 K and 35 meV at 300 K). Exfoliated $WSe_2$ monolayers exhibit comparable linewidths and have a strongly enhanced luminescence yield under ambient conditions[18], suggesting their suitability for room-temperature polaritonics. However, not even at cryogenic temperatures, strong coupling has been demonstrated in $WSe_2$ monolayers, yet.

Here, to demonstrate strong coupling at ambient conditions, we have embedded a $WSe_2$ monolayer in a compact Tamm-plasmon photonic microstructure[19,20] composed of a dielectric distributed Bragg reflector (DBR), a polymer layer and a thin gold cap. We map out the characteristic energy–momentum dispersion relations of the upper and the lower polariton branch at ambient conditions by angle-resolved PL and reflection measurements. Our experimental findings are supported by modelling our device in a coupled oscillator framework, showing an excellent agreement between theory and experiment.

## Results

**Device fabrication.** Figure 1a depicts a graphic illustration of our investigated device: it consists of a $SiO_2/TiO_2$ DBR (ten pairs), which supports a very high reflectivity of 99.97% in a wide spectral range between 580 and 780 nm. A single layer of $WSe_2$, mechanically exfoliated via commercial adhesive tape (Tesa brand) from a bulk crystal was transferred onto the top $SiO_2$ layer with a polymer stamp. The monolayer was identified by PL measurements, which also confirmed its excellent optical quality under ambient conditions (see Fig. 1b). Here we observe the characteristic profile from the A-valley exciton with a linewidth of 37.5 meV. The monolayer was capped by a 130 nm-thick layer of poly(methyl methacrylate) (PMMA) and the device was completed by a 35 nm-thick gold layer (see Supplementary Note 1 and Supplementary Figs 1 and 2 for details on the influence of PMMA on the emission properties of the monolayer). The layer thicknesses were designed to support a Tamm-plasmon resonance at the energy of the room-temperature emission energy of the A-valley exciton (1.650 eV). Figure 1c shows the vertical optical mode profile obtained by a transfer matrix calculation, the

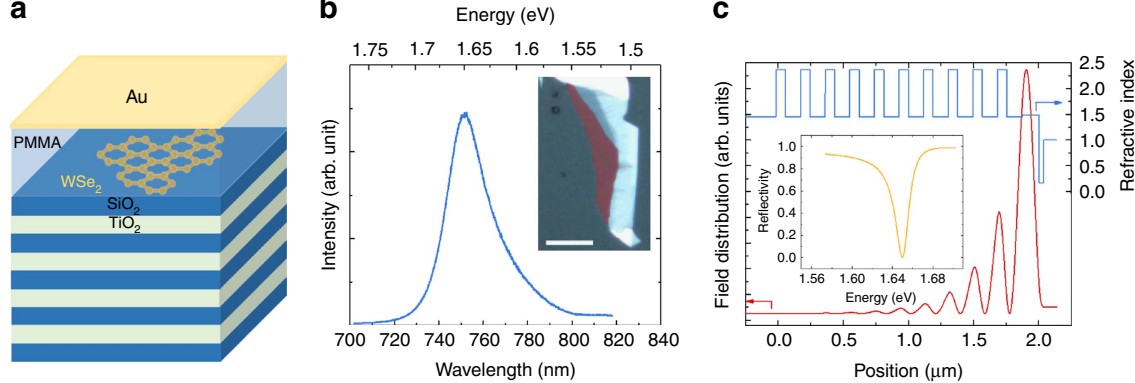

**Figure 1 | Tamm-monolayer device.** (**a**) Schematic illustration of the Tamm-plasmon device with the embedded $WSe_2$ monolayer. The monolayer is capped with PMMA, whose thickness primarily determines the frequency of the device's optical resonance. (**b**) PL spectrum of the $WSe_2$ monolayer before capping, recorded under ambient conditions. The dominant emission is identified to stem from the A-valley exciton. Inset: false-colour optical microscopy image of the used $WSe_2$ flake (monolayer in red shaded area; scale bar, 20 µm). (**c**) Calculation of the electromagnetic field intensity in the heterostructure and the optical resonance (inset). The Tamm-plasmon features a strongly enhanced field maximum close to the surface of the structure, which coincides with the vertical position of the monolayer in the device. The optical resonance features a quality factor on the order of 110.

corresponding refractive indices of the layer sequence and the resulting reflectivity spectrum (without embedded monolayer). The successful implementation of this device was confirmed by reflectivity measurements (see Supplementary Note 2 and Supplementary Fig. 3). This type of photonic microstructure features a strong field enhancement close to the metallic interface, which has proven to suffice for promoting polariton formation with embedded InGaAs-[21], GaAs-[22] and II/VI-[23] based quantum well structures at cryogenic temperatures. We point out that the large refractive index difference in the dielectric Bragg reflector leads to Tamm resonances, which provide a very strong field confinement, yet quality factors on the order of 110 can be obtained, as can be seen from the simulated reflectivity spectrum in Fig. 1c.

**Optical characterization**. In the following, we will discuss the case of a device, which contains a single layer of WSe$_2$ embedded in the resonant structure: we employ the characteristic energy–momentum dispersion relation of the vertically confined photon field in the Tamm device, to map out the coupling of the A-valley exciton of the WSe$_2$ monolayer and the Tamm-plasmon-polariton. To cover a large emission angle, thus accessing a sizeable spectral tuning range, we use a high magnification ($\times 100$) microscope objective with a numerical aperture of 0.7 in the PL experiment. As the polariton in-plane momentum $k_{||}$ is proportional to $\sin(\theta)$, with $\theta$ being the PL emission angle, this allows us to project a momentum range of up to $4.2 \, \mu m^{-1}$ onto the CCD (charge-coupled device) chip of our spectrometer in the far-field imaging configuration (see Methods section for further details). The sample is held at 300 K and the embedded monolayer is excited via a non-resonant continuous wave laser at a wavelength of 532 nm at an excitation power of 3 mW, measured in front of the microscope objective. In Fig. 2a, we plot the PL spectra extracted from our device at various in-plane momenta. At an in-plane momentum of $1.84 \, \mu m^{-1}$ (corresponding to an emission angle of 12.67°), we can observe a minimum peak distance between the two prominent features, which we identify as the lower and upper polariton branch. These two branches feature the characteristic anti-crossing behaviour with a Rabi splitting of 23.5 meV, the key signature of the strong coupling regime. We note that the strong coupling regime is primarily a result of the tight mode confinement provided by the Tamm structure[24] (see Supplementary Note 3 and Supplementary Fig. 4 for details). Figure 2b depicts the fully mapped out energy–momentum dispersion relation of the two polariton resonances by plotting the corresponding peak energies as a function of the in-plane momentum. As expected from two coupled oscillators with strongly varying effective masses, we observe the characteristic potential minimum in the lower polariton branch with a modest negative detuning of $\Delta = E_C - E_X = -11.7$ meV. This negative detuning condition leads to an effective polariton mass of $1.45 \times 10^{-5} \, m_e$ at the bottom of the lower polariton, where $m_e$ is the free electron mass. We can furthermore observe the characteristic transition from a light particle close to $k_{||} = 0$ to a heavy, exciton-like particle at large $k_{||}$ values. The corresponding Hopfield coefficients, which characterize the excitonic and photonic fraction of the lower polariton ($|X|^2$ versus $|C|^2$, respectively) are plotted as a function of the in-plane momentum in Fig. 2c. The potential minimum, which is formed in the lower polariton branch, is another key signature of an exciton polariton in the presence of vertically confined field. It furthermore provides a well-defined final energy state with a distinct effective mass, which is crucial for advanced parametric and stimulated scattering experiments[25]. A key advantage of exciton

polaritons, as compared with other composite bosons (such as excitons), is the possibility to conveniently tune the depth of this attractive potential, and simultaneously the particles' effective masses and light-versus-matter composition by changing the detuning between the light and the matter oscillator. In addition, we carried out reflectivity measurements to provide further evidence that our device works in the strong coupling regime. The results are presented in Fig. 2d and were analysed and fitted the same way as for the PL data. Owing to experimental limitations (see Supplementary Note 4) we have realized a white light reflectivity measurement with a $\times 20$ magnification objective with a numerical aperture of 0.40. Similarly, compared with the PL experiment, we observe a clear appearance of two normal modes that can be well described by a coupled oscillator model. For the sake of clarity, we have inverted the reflectivity spectra; thus, the reflection dips appear as positive signals in the graph. In Fig. 2e, we plot the extracted values of the reflection resonances as a function of the in-plane momentum, which allows us to reconstruct the polariton dispersion relation. The dispersion features the characteristic avoided crossing behaviour with a Rabi splitting of 14.7 meV. The somewhat reduced Rabi splitting measured in reflectivity is expected for a photonic structure with a modest quality factor and the slightly shifted resonance of the excitonic oscillator is most likely to be resulting from sample ageing, which is both detailed in the Supplementary Note 4.

**Modelling**. To interpret our experimental data, we can fit the dispersions with a coupled oscillator model:

$$\begin{bmatrix} E_{ph} + i\hbar\Gamma_{ph} & \hbar\Omega/2 \\ \hbar\Omega/2 & E_{ex} + i\hbar\Gamma_{ex} + \Delta \end{bmatrix} \begin{bmatrix} \alpha \\ \beta \end{bmatrix} = E \begin{bmatrix} \alpha \\ \beta \end{bmatrix} \quad (1)$$

where $E_{ph}$ and $E_{ex}$ are photon and exciton energies, respectively, $\Delta$ is the detuning between the two modes, and $\Gamma_{ph}$ and $\Gamma_{ex}$ are photon and exciton mode broadening, respectively. The eigenvectors represent the weighting coefficients of exciton and photon fraction and $\hbar\Omega$ represents the Rabi splitting in the system. Solving the dispersion equation:

$$det \begin{bmatrix} E_{ph} + i\hbar\Gamma_{ph} - E & \hbar\Omega/2 \\ \hbar\Omega/2 & E_{ex} + i\hbar\Gamma_{ex} + \Delta - E \end{bmatrix} \begin{bmatrix} \alpha \\ \beta \end{bmatrix} = 0 \quad (2)$$

one can obtain two polariton branches. The result of this modelling is shown in Fig. 2b,e (solid lines) along with the experimental data (symbols). The fitting was carried out via solving the optimization problem with detuning, Rabi splitting and photon mass used as parameters. As the exciton mass is several orders of magnitude larger than the photon mass, it does not affect the result of the simulation and its value is taken to be 0.8 $m_e$, as defined in ref. 26. The dashed lines show photon and exciton energies as a function of the in plane wave vector $k_{||}$. For details on the fitting procedure of momentum-resolved spectra see Supplementary Note 5 and Supplementary Figs 5 and 6.

We will now address the occupation of the polariton states in our device, operated under ambient conditions. The overall, momentum-resolved PL spectrum of the structure is plotted in Fig. 3a. In stark contrast to previous reports discussing polariton emission with TMDC materials at room temperature[16], we observe a pronounced occupation of the low-energy states in the lower polarion branch and a reduced occupation of the excited polariton states. The following model was used to analyse the luminescence experiment: in a first approximation, owing to the comparably low particle numbers and high temperatures, we assume a Boltzmann distribution law for our particles: $N_i \sim \exp(-E_i/k_B T)$, where $N_i$ and $E_i$ denote $i$-state population

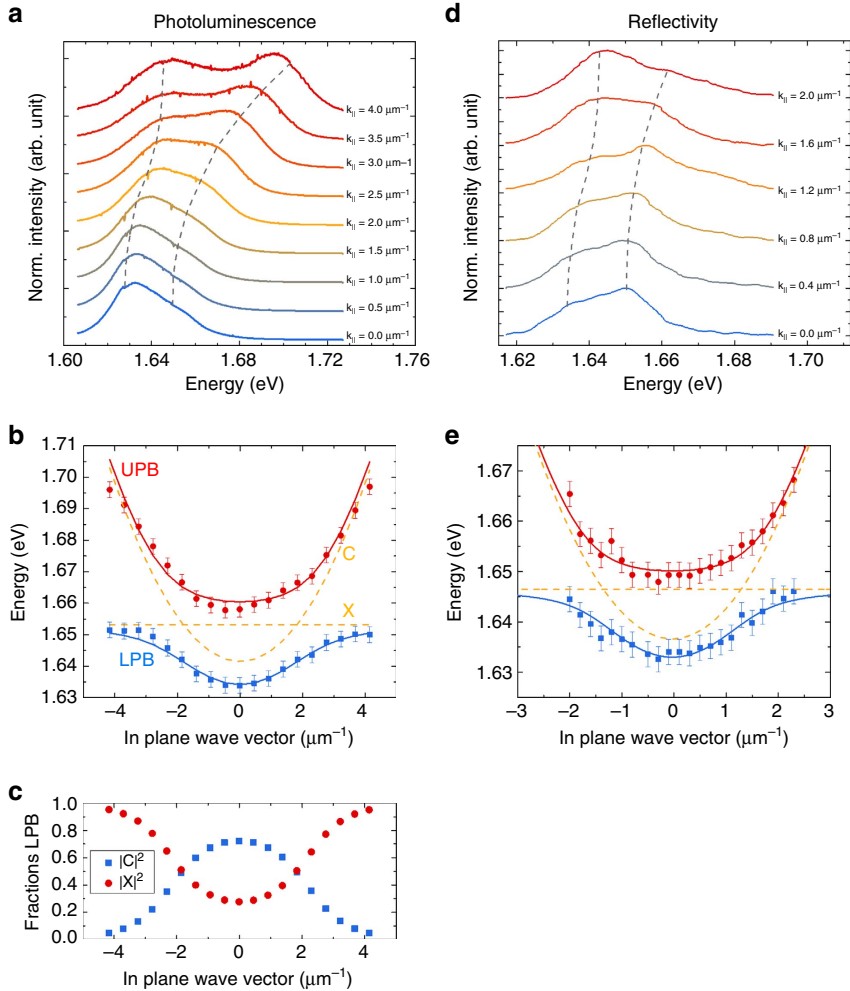

**Figure 2 | Exciton-polariton formation with Tamm-plasmons.** (**a**) PL spectra recorded from the coupled device at room temperature at various in-plane momenta (depicted in a waterfall representation). Two pronounced resonances evolve in the system, which feature the characteristic anti-crossing behaviour of exciton-polaritons. (**b**) Energy–momentum dispersion relation of the lower and upper polariton branch at room temperature: the polariton energies are extracted by fitting spectra at various in-plane momenta (solid symbols). A coupled oscillator approach is employed to fit the data and to demonstrate excellent agreement between experiment and theory (lines). (**c**) Plot of the exciton and photon fraction of the lower polariton branch as a function of the in-plane momentum extracted from coupled oscillator fit. (**d**) Inverted reflectivity spectra at different in-plane momenta. (**e**) Energy–momentum dispersion relation extracted from the reflectivity spectra.

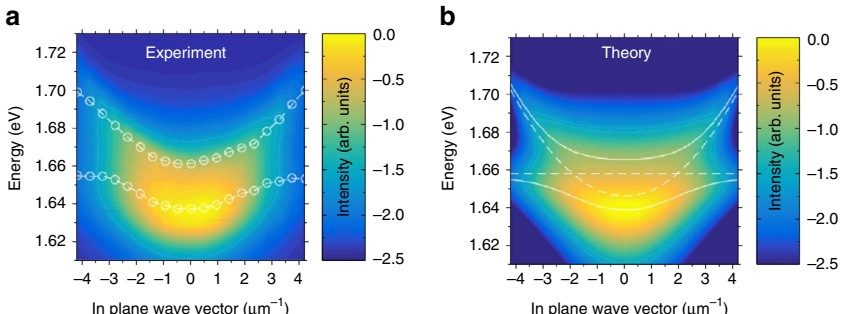

**Figure 3 | Experimental and theoretical polariton dispersion relations in the studied Tamm structure.** (**a**) Room-temperature false colour intensity profile of the full polariton dispersion relation extracted from the PL measurements. (**b**) Model of the full dispersion by assuming a Boltzmann distribution of the quasiparticles with an effective temperature of 300 K.

and energy, and $k_B$ is the Boltzmann constant. The modelled PL is thus generated by a polariton gas at room temperature ($T = 300$ K). We further assume that the emission stems from the photonic mode only and is broadened in energy according to a Lorentz distribution. This allows us to relate the PL intensity to the photonic Hopfield coefficients via:

$$I(k, E) \sim \sum_i \frac{\left| C_{ph}^i \right|^2 \exp\left( -E_i(k)/k_B T \right)}{(E - E_i(k))^2 + \Gamma_{ph}^2} \quad (3)$$

where $\Gamma_{ph}$ is the broadening of the photonic mode and the $i$-index spans over the two polariton branches. We extract the value of $\Gamma_{ph} = 15$ meV from the experimental data. The experimental results and the theoretically calculated dispersion relation are plotted in Fig. 3a,b, respectively.

In fact, we achieve very good agreement between theory and experiment. Although our model is purely phenomenological and cannot account for any dynamic and microscopic effects in our system, it already serves as a first indicator that, despite the pronounced dissipation in our system, polariton relaxation is indeed significant. Additional simulations of our system with higher temperatures, which result in stronger luminescence from the upper polariton branch, can be found in Supplementary Note 6 and Supplementary Fig. 7 of the manuscript.

## Discussion

In conclusion, we have observed clear evidence for the formation of exciton polaritons in a hybrid dielectric and polymer Tamm-plasmon-polariton device featuring an integrated single atomic layer of the TMDC WSe$_2$. We mapped out the distinct polariton dispersion relation in angle-resolved PL and reflectivity measurements, and resolved both polariton branches including the characteristic parabolic energy minimum and the flattening towards the exciton band. Our experimental data are supported by a coupled harmonic oscillator model and we achieve very good agreement both for the energy evolution of the polariton resonances and for the population of polariton eigenstates. We believe that our work represents a significant step towards the implementation of polariton condensates and nonlinear experiments in the strong coupling regime based on single layers or stacks of several layers of TMDCs. Moreover, it will be of particular interest for TMDC polaritonic experiments to harness the unique spinor and valley physics inherited by the atomic monolayers. Combining plasmonics[27,28] and two-dimensional active media in the strong light–matter coupling regime certainly carries great potential for building new architectures of highly integrated, nonlinear optical circuits and logic devices, which are operated at ultra-low powers and close to terahertz frequencies.

## Methods

**Sample design and fabrication.** The sample was designed based on transfer matrix calculations, where the plasmon-polariton resonance was tuned to match the A exciton resonance of WSe$_2$ monolayer at room temperature (1.650 eV). The bottom mirror consists of a commercially available DBR based on a fused silica substrate topped with ten pairs of TiO$_2$/SiO$_2$ layers (72/117 nm thickness, respectively, corresponding to a central stopband wavelength of 680 nm). The stopband ranges from 580 to 780 nm. The WSe$_2$ monolayer was mechanically exfoliated onto a polymer gel film (polydimethylsiloxane) and was then transferred onto the bottom DBR. One hundred and thirty nanometres of PMMA were deposited by spin coating onto the structure. Finally, a 35 nm-thick gold layer was thermally evaporated onto the sample.

**Experimental setup.** We took advantage of an optical setup in which both spatially (near-field) and momentum-space (far-field)-resolved spectroscopy and imaging are accessible. PL is collected through a 0.7 numerical aperture microscope objective lens and directed into an imaging spectrometer with 1,200 groves mm$^{-1}$ grating via a set of relay lenses, projecting the proper projection plane onto the monochromator's entrance slit. The system's angular resolution is $\sim 0.05\,\mu m^{-1}$ ($\sim 0.5°$) and spectral resolution is $\sim 0.050$ meV with a nitrogen-cooled Si-CCD as detector.

**Analysis of the PL spectra.** As the fine spectrometer grating (1,200 groves mm$^{-1}$) does not cover the full spectral range of the PL signal, the angle-resolved PL spectra were taken at three different energies with $\sim 60\%$ overlap and were subsequently combined. This procedure used a fast Fourier transform (FFT) smoothing algorithm to account for small-intensity offsets in the overlap regions. Finally, single spectra were fitted with a two-Gaussian fit, to deduce the peak energy.

**Data availability.** The data that support the findings of this study are available from the corresponding author upon request.

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

## Acknowledgements

This work has been supported by the State of Bavaria. We thank S. Tongay for providing initial samples for the project. We further thank S. Stoll and I. Kim for assistance in the exfoliation and pre-characterization of the high-quality monolayer material. Moreover, we thank A. Wolf for fabrication assistance and S. Brodbeck for assistance with the transfer matrix calculations. C.S. thanks L. Worschech for encouraging him at the very early stage of the work. A.K. and S.H. acknowledge the partial financial support from the EPSRC Hybrid Polaritonics Programme. C.S. acknowledges financial support by the European Research Council (unLiMIt-2D project).

## Author contributions

C.S and S.H. initiated the study and guided the work. N.L., C.P.D. and C.S. designed the Tamm device. N.L. and O.I. exfoliated, identified and transferred the monolayer. N.L. fabricated the Tamm structure. N.L., S.K. and M.K. performed experiments. N.L., S.K., S.B. and C.S. analysed and interpreted the experimental data, supported by all co-authors. E.C., A.V.N. and A.V.K. provided the theory. C.S., N.L., S.K. and E.C. wrote the manuscript, with input from all co-authors.

## Additional information

**Competing financial interests:** The authors declare no competing financial interests.

**Publisher's note**: 

