## [Peer Review File · Nature Communications]

Reviewers' comments:

Reviewer #1 (Remarks to the Author):

A. The main results include observation of strong exciton-photon coupling between Tamm plasmons and excitons in monolayer of WSe₂.

B. The main motivation in studying exciton-polaritons is potential application of this high nonlinear system in the development of all-optical signal processing devices. Although strong coupling has been already reported with MoSe₂ and MoS₂, the novelty of this work is the observation of strong coupling with another material WSe₂ and in another geometry (Tamm plasmon). Also the experiments are done at 300 K, whereas strong coupling with MoSe₂ was observed only at 4 K.

C.D The experimental data presented are in good agreement with theoretical model. My only concern is that the linewidths of polariton mode are quite broad and as a result the lower and upper polaritons are not really resolved. Anyway the observed angular-resolved PL clearly shows that PL consists of a doublet, which shows an anticrossing behaviour.

E.F. I think the conclusions are supported by the experimental observation. My only suggestion would be to show data at low temperature of 4 K and how polariton linewidths change with reduction of T from 400 to 4 K. At 4K the doublet of lower and upper polariton states should be clearly resolved.

G.H. All fine.

Reviewer #2 (Remarks to the Author):

The manuscript demonstrates the realization of microcavity structure embedded with monolayer WSe₂ and the characterization of momentum space photoluminescence (PL) at room temperature. Based on the experimental analysis and simulation modeling, the manuscript claims the formation of exciton polaritons at room temperature and the thermally distributed polariton emission via Boltzmann fitting. This metal-DBR microcavity structure is an effective alternative structure to show the strong exciton-photon coupling in 2D materials. However, there are some issues in the evidence, which is insufficient to support the claim. Thus I cannot recommend the acceptance of this manuscript unless some major revisions are considered.

Major concerns:

- 1) For the strong coupling demonstration, only momentum space PL is demonstrated at room temperature. Reflectivity or transmission spectra in the momentum space are needed to confirm the strong coupled polariton modes.
- 2) Generally at different dielectric environments, the PL emission of 2D materials could be either exciton-dominant, or trion-dominant, or even defect-dominant, where the linewidth could also become different due to some inhomogeneous broadening. At room temperature, the WSe₂ sample without capping layer (Fig. 1b) shows exciton-dominant emission (37.5 meV), which does not guarantee the same situation after the capping layer is deposited. The PL is needed to be confirmed as consistent after the capping layer.
- 3) No control sample has shown the exciton temperature-dependence. Because of the variation of the exciton emission and temperature-dependence due to different dielectric environments and substrate strains, the exciton energy at 260K determined to be at $\sim 1.665\text{eV}$ seems to be based on no evidence at this manuscript. If no exciton temperature-dependence are shown on this specific scheme, the description and analysis of the PL at 260K is not convincing.
- 4) The "polariton dispersion" at 260 K has large discrepancy with the coupled oscillator fitting as in Fig. 2c. And the lowest state of the upper polariton is clearly lower than the exciton line and the

highest lower polariton modes, showing a crossing feature. Are the trions becoming important at this temperature? Is there any linewidth broadening at this temperature? More experiments or some explanation or are needed to demonstrate if at this temperature the system is still as comparable to that at room temperature, and why it is so.

5) As in the manuscript, the polariton emission is modeled as Boltzmann distribution as the low particle number of polaritons is assumed. The Boltzmann distribution of polaritons is validated when the thermal equilibrium has reached (Kasprzak et al, Nature (2006), 443, 409). The thermal equilibrium in the monolayer TMD exciton-polariton strongly depends on the dynamics of TMD excitons since the TMD exciton dynamics is fast as less than 10ps, which is much faster than most of the conventional inorganic quantum wells. This lifetime scale generally could become even smaller if any of quenching mechanisms happens from the dielectric environment. With such fast dynamics in comparison with thermalization time, the thermal equilibrium of the polaritons is not so likely. If no dynamics evidence is shown to support the thermal equilibrium, the Boltzmann modeling and the thermal polariton temperature need to be carefully reconsidered.

Other revisions:

1) Are the assignments of the peaks in the PL dispersion based on the multi-peak fitting (Gaussian or Lorentzian)? Based on the extracted PL spectra, it is hard to determine all the peak positions. If based on the fitting, specific fitting example should be clarified, and reasonable error bars need to be added to the dispersion data.

2) Which wavelength is simulated for the electric field in Fig. 1c? Is it at the exciton resonance or the cavity resonance? The designed electric field maximum of this cavity is located in the middle of capping PMMA layer, not at the interface between DBR and the PMMA layer. Careful thickness control of the top SiO₂ layer of DBR and PMMA layer could move the electric field maximum to overlay on the monolayer WSe₂.

3) Is the quality factor of 110 on page 4 based on the simulation data of Fig. 1c or the Fig. S1a? Is it more suitable to put the experimental reflectivity at Fig. 1c?

4) How large is the pump beam spot size? Only the central area of the monolayer flake shows about 5μm width. Please confirm the pump beam spot is small enough within the monolayer flake and elaborate it in the caption or main text.

5) There are some typos in the references, and it is not with the right format and journal abbreviations.

Reviewer #3 (Remarks to the Author):

The authors report on observation of room temperature polaritons in a WSe₂ monolayer inserted in a two-dimensional microcavity with the bottom mirror formed by a dielectric DBR and the top metallic mirror deposited on the polymer cavity.

This paper is an important step towards observation of non-linear polariton phenomena at room temperature. It also shows that relatively uncomplicated microcavity design can be used for two-dimensional microcavities comprising TMDC layers, which is very promising for the development of this research field.

Overall the paper merits publication in Nature Communications. However, there are several points that need to be addressed and thoroughly revised.

1. The exciton linewidth in WSe₂ at room T is typically about 40 meV. There are indications that this maybe the value of a homogeneous linewidth. This is larger than the Rabi splitting claimed in this work. According to the text book definition (see Kavokin, Baumberg, Malpuech, Laussy 'Microcavity' book) the Rabi splitting should be larger than the difference of the exciton and cavity linewidths, the condition not fulfilled in this work. It would be important that authors comment on this, and explain why that condition is unimportant for their claim. Room temperature PL

measurements on bare WSe2 monolayer would be very helpful to show.

2. It would really help if the peak fitting procedure for the spectra in Fig.2a is explained and illustrated in detail (either in the main text or SI). Perhaps the measured spectra could be shown together with the fitting on the same graph.

3. The importance of the 'plasmonic' properties of the device is somewhat exaggerated. This is especially striking in the abstract where it is mentioned that 'plasmonic architectures... is a crucial step towards compact... photonic and polaritonic circuits', and then raised in more places in the text. It is not clear what plasmonic effects the authors refer to, and what importance they bare for the reported observation of the strong coupling. I understand that there is a body of work on Tamm polaritons, but emphasis on plasmonic effects is nonetheless misleading. This should be thoroughly revised.

In the following document, we provide detailed answers to the referees' comments

Reviewer #1 (Remarks to the Author):

A. The main results include observation of strong exciton-photon coupling between Tamm plasmons and excitons in monolayer of WSe2.

B. The main motivation in studying exciton-polaritons is potential application of this high nonlinear system in the development of all-optical signal processing devices. Although strong coupling has been already reported with MoSe2 and MoS2, the novelty of this work is the observation of strong coupling with another material WSe2 and in another geometry (Tamm plasmon). Also the experiments are done at 300 K, whereas strong coupling with MoSe2 was observed only at 4 K.

C.D The experimental data presented are in good agreement with theoretical model. My only concern is that the linewidths of polariton mode are quite broad and as a result the lower and upper polaritons are not really resolved. Anyway the observed angular-resolved PL clearly shows that PL consists of a doublet, which shows an anti-crossing behavior.

E.F. I think the conclusions are supported by the experimental observation. My only suggestion would be to show data at low temperature of 4 K and how polariton linewidths change with reduction of T from 400 to 4 K. At 4K the doublet of lower and upper polariton states should be clearly resolved.

Authors:

We thank the reviewer for his/her interesting suggestion. Such an experiment could indeed shed a lot of light into the physics of polaritons emerging in 2D materials. However, we want to point out that our Tamm structure does not provide any way of detuning the optical mode, which would be required to compensate for the temperature-induced energy shift of the exciton energy. At room temperature the detuning is already moderately negative. At 4 K the detuning between

exciton and optical mode would amount to -100 meV (see supplementary information S5, added in the resubmitted version of the manuscript), which makes the observation of strong coupling signatures impossible.

G.H. All fine.

Reviewer #2 (Remarks to the Author):

The manuscript demonstrates the realization of microcavity structure embedded with monolayer WSe₂ and the characterization of momentum space photoluminescence (PL) at room temperature. Based on the experimental analysis and simulation modeling, the manuscript claims the formation of exciton polaritons at room temperature and the thermally distributed polariton emission via Boltzmann fitting. This metal-DBR microcavity structure is an effective alternative structure to show the strong exciton-photon coupling in 2D materials. However, there are some issues in the evidence, which is insufficient to support the claim. Thus I cannot recommend the acceptance of this manuscript unless some major revisions are considered.

Major concerns:

1) For the strong coupling demonstration, only momentum space PL is demonstrated at room temperature. Reflectivity or transmission spectra in the momentum space are needed to confirm the strong coupled polariton modes.

Authors:

We thank the referee for this suggestion. Indeed reflectivity measurements to map out polariton resonances are rather standard in more conventional polariton cavities and devices (GaAs, II-VI, organics). However, in our case, we are dealing with a Tamm-plasmon structure, which does not feature a comparably pronounced reflection minimum such as a symmetric Fabry-Perot cavity. This inherently results in weaker reflection signals. Second, and more important, the size of our monolayer is limited, which makes a very high spatial resolution necessary while preserving a reasonable angular resolution. While this is possible in the photoluminescence configuration, it is extremely demanding in reflection measurements, as increasing spatial resolution always comes to the expense of a strong decrease in the signal to noise ratio. We have nevertheless carried out this cumbersome experiment, and could successfully reproduce the energy momentum dispersion relation of the upper and lower polariton branches at ambient conditions.

We believe that these data serve as a complementary evidence that our device works in the strong coupling regime. We have implemented these data in supplementary section S8, including a short discussion:

Figure R1: left, inverted reflectivity spectra at different in-plane momenta. right, polariton dispersion relation based on a two-Gaussian fit of the reflectivity data.

Due to experimental limitations discussed above we have realized a white light reflectivity measurement with a 20x magnification objective with a numerical aperture of 0.40. (Measurements with a higher NA=0.7 objective as for the PL to access higher k -values have not been possible, again see explanation above). The results in comparison to the photoluminescence data can be seen in Fig. R1/S8. As in the PL we observe a clear appearance of two normal modes that can be well described by a coupled oscillator model. The somewhat reduced Rabi splitting measured in reflectivity is expected and in good, quantitative agreement with Savona et al., *Solid State Commun.* **93**, 733 (1997), p. 8-9, eqn. (17)-(21).

2) Generally at different dielectric environments, the PL emission of 2D materials could be either exciton-dominant, or trion-dominant, or even defect-dominant, where the linewidth could also become different due to some inhomogeneous broadening. At room temperature, the WSe₂ sample without capping layer (Fig. 1b) shows exciton-dominant emission (37.5 meV), which does not guarantee the same situation after the capping layer is deposited. The PL is needed to be confirmed as consistent after the capping layer.

Authors:

We agree with the referee that dielectric environment can generally affect the PL emission. In fact, we had checked the influence of PMMA on the PL emission prior to our investigations. We found that the peak energy is slightly blue shifted and the intensity and linewidth are not significantly affected. Therefore, we can exclude that the PL emission changes to trionic or even defect-dominated emission due to the PMMA capping. In order to clarify this for the reader, we have added supplementary information S4, which reads as follows:

“Here, we present the influence of PMMA on the PL emission of WSe₂ monolayers. Figure S5 shows the PL spectra of separate WSe₂ monolayer that was measured before and after PMMA capping. PL setup adjustments and excitation power (70 μW) were kept the same for both measurements. The peak energy shifts blue by 12 meV and the linewidth decreases from 42.1 meV to 38.7 meV. The slightly larger linewidth of this flake is due to the slightly varying optical quality of the individual monolayers. It should be also noted that the intensity is hardly affected by the capping at all.”

Figure S5 | Photoluminescence spectra of a WSe₂ monolayer at room temperature before and after capping with PMMA

3) No control sample has shown the exciton temperature-dependence. Because of the variation of the exciton emission and temperature-dependence due to different dielectric environments and substrate strains, the exciton energy at 260K determined to be at $\sim 1.665\text{eV}$ seems to be based on no evidence at this manuscript. If no exciton temperature-dependence are shown on this specific scheme, the description and analysis of the PL at 260K is not convincing.

Authors:

We thank the referee for his suggestion and agree that the exciton temperature dependence is helpful for the reader. We have used this dependence, now presented in the supplementary information S5, in order to determine the sample temperature in the 260 K case. S5 reads as follows:

“This supplementary section provides information on the temperature-induced shift of the exciton energy. Figure S6 shows the energy-temperature dependence of the PL of a separately measured WSe₂ monolayer. Although, the absolute peak energy may vary slightly depending on dielectric environment or strain condition, the measured slope of 0.34 meV/K can be used to calibrate the sample temperature.”

Figure S6|: Exciton energy as a function of temperature

4) The "polariton dispersion" at 260 K has large discrepancy with the coupled oscillator fitting as in Fig. 2c. And the lowest state of the upper polariton is clearly lower than the exciton line and the highest lower polariton modes, showing a crossing feature. Are the trions becoming important at this temperature? Is there any linewidth broadening at this temperature? More experiments or some explanation or are needed to demonstrate if at this temperature the system is still as comparable to that at room temperature, and why it is so.

Authors:

We would like to emphasize, that there is no 'crossing' feature in the data. Both the upper and lower polariton branch clearly feature the avoided crossing behavior, and in addition, the lower polariton branch almost perfectly follows the polariton dispersion relation. We would like to emphasize, that the exciton resonance is not a sharp line but a peak with a linewidth of 35 - 40 meV, thus it is not too surprising that some of our experimental data cross this fit by a few meV.

In the revised manuscript, we have added a careful estimation of the experimental errors: The data of the lower polariton match perfectly the coupled oscillator fit within the errors. In the upper branch, the data at low k values and very large k values still deviate from the fit by a few meV (which is significantly less than their linewidth), yet no crossing is evident within the experimental accuracy.

While it is reasonable to exclude strong contributions from trions, polarons, biexcitons and defects at room temperature (both are unstable under ambient conditions), we cannot completely rule out slight contributions of trions to the photoluminescence at 260 K. The emission feature, which is commonly attributed to trions (or attractive polarons) in WSe₂ could be expected on the low energy side of the excitonic resonance. Previous works by You et al. Nature Physics (2015) also have suggested the presence of a second resonance (assigned to a biexciton) in WSe₂, which emerges on the low energy side of the exciton. The contribution of such additional oscillators indeed might explain the deviation of the upper polariton from the two oscillator fit. However, in our opinion, the inclusion of various oscillators without a precise knowledge of their respective oscillator strength would not significantly shed more light into the physics of our device, as it primarily would increase the number of free fitting parameters. Thus, we would like to keep the current representation of our data.

5) As in the manuscript, the polariton emission is modeled as Boltzmann distribution as the low particle number of polaritons is assumed. The Boltzmann distribution of polaritons is validated when the thermal equilibrium has reached (Kasprzak et al, Nature (2006), 443, 409). The thermal equilibrium in the monolayer TMD exciton-polariton strongly depends on the dynamics of TMD excitons since the TMD exciton dynamics is fast as less than 10ps, which is much faster than most of the conventional inorganic quantum wells. This lifetime scale generally could become even smaller if any of quenching mechanisms happens from the dielectric environment. With such fast dynamics in comparison with thermalization time, the thermal equilibrium of the polaritons is not so likely. If no dynamics evidence is shown to support the thermal equilibrium, the Boltzmann modeling and the thermal polariton temperature need to be carefully reconsidered.

Authors:

We thank the referee of his remark. We fully agree with the referee that considering the inherent timescales of the system a full thermalization of the exciton-polaritons seems somewhat unlikely. In addition, whether the mechanism responsible for the distribution of polaritons is thermal or driven-dissipative is not well established, as thermal-like Boltzmann tails in the polariton spectrum can be sometimes coincidental or can result from quantum noise of purely driven-dissipative origin. (see Bajoni et al. Phys. Rev. B 76, 201305(R) (2007), A. Chiochetta and I. Carusotto, Phys. Rev. A 90, 023633 (2014)).

We emphasize that giving a reliable conclusion on the thermalization of polaritons in a WSe₂ layer at room temperature is out of the scope of our present work. Interestingly enough, a somewhat naive, yet intuitive first approximation of a Boltzmann-like distribution results in a very good agreement of theory and experiment. The microscopic origin of this mechanism is to be investigated in future works. In the present version of the manuscript we have included the results of Boltzmann modelling performed assuming the effective temperatures of the polariton gas ranging between 300 and 1500 K in the supplementary information S6. The best fit to the experiment is achieved at the effective temperature between 300 and 500 K that shows that the

polariton gas is out of equilibrium with the crystal lattice indeed. A more accurate calculation would require the knowledge of polariton-polariton and polariton-phonon interaction constants. To avoid to possibly mislead the reader we have slightly modified the main text:

“In fact, we achieve very good agreement between theory and experiment. While our model is purely phenomenological and cannot account for any dynamic and microscopic effects in our system, it already serves as a first indicator that despite the pronounced dissipation in our system, polariton relaxation is indeed significant. Additional simulations of our system with higher temperatures, which result in stronger luminescence from the upper polariton branch, can be found in the supplementary section of the manuscript.”

We have added a new paragraph in the supplementary material section where we compare different effective temperatures in the framework of our model.

Other revisions:

1) Are the assignments of the peaks in the PL dispersion based on the multi-peak fitting (Gaussian or Lorentzian)? Based on the extracted PL spectra, it is hard to determine all the peak positions. If based on the fitting, specific fitting example should be clarified, and reasonable error bars need to be added to the dispersion data.

Authors:

We have included fitting examples in the supplement S3 of the revised manuscript, as demanded by the referee. The assignments of the peaks indeed is based on double Gaussian fitting. In the methods section we have changed “two-peak fit” to “double-Gaussian fit”.

2) Which wavelength is simulated for the electric field in Fig. 1c? Is it at the exciton resonance or the cavity resonance? The designed electric field maximum of this cavity is located in the middle of capping PMMA layer, not at the interface between DBR and the PMMA layer. Careful thickness control of the top SiO₂ layer of DBR and PMMA layer could move the electric field maximum to overlay on the monolayer WSe₂.

Authors:

The simulation has been carried out for the wavelength equivalent to the exciton resonance. As the thickness of the DBR stacks does not perfectly match the ideal values, the thickness of the PMMA layer has been adjusted to match the spectral resonance between the monolayer and the photonic device. As a consequence, the monolayer is in fact not exactly situated at the position of the field maximum. Thus, fig. 1c illustrates a realistic estimate on field distribution and monolayer position.

3) Is the quality factor of 110 on page 4 based on the simulation data of Fig. 1c or the Fig. S1a? Is it more suitable to put the experimental reflectivity at Fig. 1c?

Authors:

The experimental Q factor ($Q = 110$) matches the theoretical Q factor ($Q = 112$) within the measurement and simulation error. Therefore, we use the formulation “on the order of 110”. In this case, we would like to keep the current presentation.

4) How large is the pump beam spot size? Only the central area of the monolayer flake shows about 5 μ m width. Please confirm the pump beam spot is small enough within the monolayer flake and elaborate it in the caption or main text.

Authors:

A conservative estimate of the pump spot (excitation with a 100X magnification objective, 0.7 NA) yields a value of 2 μ m. Given the flake size (see figure 1b), we are certain that the excitation beam spot is significantly smaller than the monolayer.

5) There are some typos in the references, and it is not with the right format and journal abbreviations.

Authors:

We have carefully checked the references, and hope that we do now match the guidelines.

Reviewer #3 (Remarks to the Author):

The authors report on observation of room temperature polaritons in a WSe₂ monolayer inserted in a two-dimensional microcavity with the bottom mirror formed by a dielectric DBR and the top metallic mirror deposited on the polymer cavity.

This paper is an important step towards observation of non-linear polariton phenomena at room temperature. It also shows that relatively uncomplicated microcavity design can be used for two-dimensional microcavities comprising TMDC layers, which is very promising for the development of this research field.

Overall the paper merits publication in Nature Communications. However, there are several points that need to be addressed and thoroughly revised.

1. The exciton linewidth in WSe₂ at room T is typically about 40 meV. There are indications that this maybe the value of a homogeneous linewidth. This is larger than the Rabi splitting claimed in this work. According to the text book definition (see Kavokin, Baumberg, Malpuech, Laussy 'Microcavity' book) the Rabi splitting should be larger than the difference of the exciton and cavity linewidths, the

condition not fulfilled in this work. It would be important that authors comment on this, and explain why that condition is unimportant for their claim. Room temperature PL measurements on bare WSe2 monolayer would be very helpful to show.

Authors:

*We thank the referee for his comment on this important point. In fact, the key criterion to observe strong coupling conditions is the anti-crossing of the two resonances, while the linewidth of the coupled oscillators can exceed the Rabi-splitting. Indeed, our splitting is on the order of the difference between the linewidths. As shown by Savona et al. Solid State Commun. **93**, 733 (1997) in the case of a cavity with a modest reflectivity the observed splitting in PL (Ω_{PL}) and reflectivity (Ω_R) systematically underestimate the real Rabi splitting (Ω). This is observed by us in a difference in the splitting between PL and reflectivity that is presented in Fig. R1/S8, which is in good agreement with Savona et al.*

While we agree with the referee that our structure is at the edge of where strong coupling can be observed, we are confident that the presence of strong coupling is unambiguously demonstrated in our work.

2. It would really help if the peak fitting procedure for the spectra in Fig.2a is explained and illustrated in detail (either in the main text or SI). Perhaps the measured spectra could be shown together with the fitting on the same graph.

Authors:

We have added a detailed peak fitting procedure, as suggested by the referee, in the supplementary information S3.

3. The importance of the 'plasmonic' properties of the device is somewhat exaggerated. This is especially striking in the abstract where it is mentioned that 'plasmonic architectures... is a crucial step towards compact... photonic and polaritonic circuits', and then raised in more places in the text. It is not clear what plasmonic effects the authors refer to, and what importance they bare for the reported observation of the strong coupling. I understand that there is a body of work on Tamm polaritons, but emphasis on plasmonic effects is nonetheless misleading. This should be thoroughly revised.

Authors:

We agree with the referee and modified the abstract and introduction accordingly to place our work in a more polariton-oriented context.

In the following, we provide detailed answers to the referee's queries.

1) As the authors suggested, it is understandable that it is difficult for reflectivity measurements on this small 2D flake. But the reflectivity dispersion directly represents the new eigenstates of polaritons in the photonic band structure (as the absorption modes) while the PL measurements can only be able to clearly represent the polariton unless no complicated polariton relaxation process is involved (which is generally not the case for very fast decay excitonic systems). Thus it cannot be only taken as the complementary evidence, but should be as the primary evidence in supporting the claim of strong coupling regime.

Our response: We thank the referee for this comment. In the revised version of the paper, we are showing the reflectivity spectra in fig. 2d and 2e, respectively, to support our strong coupling claim. One can clearly see the characteristic anticrossing features in the reflectivity.

2) For the reflectivity measurements, there are some serious issues which make the claimed "strong coupling regime" not so likely.

i) The exciton line as in Fig. S7 is 1.646 eV, but the exciton in Fig. 2b is ~ 1.654 eV. The ~ 8 meV exciton energy difference for the fitted Rabi splitting (claimed 14 meV in reflectivity, 23.5 meV in PL) is very significant. If the Fig. S5 "after PMMA" energy (~ 1.654 eV) is taken as the reference exciton line, it will just make a clear crossing feature of the upper branch in Fig. S7.

Our response: The difference of the exciton lines is explained by the Stokes shift between absorption and PL emission. In fact, the Stokes shift in our experiment amounts to 6.8 meV. This value is fully in line with the current literature value of 5 meV (see Yan et al. APL 105 (2014))

ii) As in the strong coupling regime, these two splittings from reflectivity and PL would be quite close. The surprising reduced Rabi splitting from 23.5 meV to 14 meV, similar to the theoretical reference (Savona et al., Solid State Commun. 93, 733) comparing the difference of PL and reflectivity dispersions, is very consistent with the so called "intermediate coupling regime" in the later review (Houdre, R., Physica Status Solidi (B), 242(11), 2167 (2005), Fig. 13 and Fig. 14). In this regime, there will be "normal mode splitting" in the PL (or transmission) but just one mode in the reflectivity (or absorption).

Our response: We thank the referee for carefully considering possible regimes of our system:

First: for a system where reflectivity (Q-factors) is modest (which is the case in our device), a deviation between PL and Reflectivity is not surprising. It is expected indeed following the well-known theoretical works (see Savona et al). The reduction of the Rabi splitting in Reflectivity is in full quantitative agreement with the expressions obtained by Savona et al for a coupled oscillator in the strong coupling regime.

In contrast, in the case of 'intermediate coupling', as the referee points out, reflectivity would be a single mode. One can clearly see in fig 2d of the revised manuscript (and previously fig. S8), our reflection spectra feature two resonances with a clear avoided crossing behavior. This allows us ruling out the 'intermediate coupling' regime and confirms that the strong coupling regime is established in our system.

3) As the other two reviewers pointed out the issue of the linewidths, I also carefully examined the linewidth in this work. The direct observation of full width half maximum (FWHM) from the spectrum in Fig. 1b is ~ 25 nm (~ 50 meV, and similar FWHM is also observed in Fig. S5) while the claimed linewidth in the main text is 37.5 meV (extracted from some peak fitting to exclude the trion contribution?).

Our response: The best fit of the spectrum is achieved assuming an oscillator with a linewidth of 37.5 meV. The spectrum features an asymmetry, which is accounted for by including the second oscillator of lower oscillator strength and low occupancy (it looks like a weak trionic feature). To reveal the microscopic origin of this oscillator would require a significant supplementary experimental study that is beyond the scope of this work.

4) For the criterion of the strong coupling regime, the Rabi splitting in the reflectivity (14 meV) with the linewidths, on the other way than PL, does not support the strong coupling regime.

Our response: Actually, the figure of ~ 14 meV is not the "bare" splitting of exciton and polariton modes as if there is no broadening, but the real splitting that accounts for the broadening of both exciton and photon.

It is given by a square root of the squared exciton-photon coupling energy ("bare" Rabi energy) minus a term which accounts for the broadening of the oscillators. The latter term differs for reflection and PL (see e.g. the Savona paper Solid State Commun. 93, 733). Once this value is positive, the system is certainly in the strong coupling regime. Hence, both the reflectivity and PL data support our claim for the strong coupling.

5) The explanation of Fig. 2c (reflectivity dispersion at 260K) is not convincing. The upper branch at small k-inplane is definitely below the exciton line, which is a crossing feature. If looking closer to the dispersion data points of Fig. 2c, the two branches with "parallel" curvatures look like two parallel photonic modes, deviating from the upper branch fitting curve up to more than 10 meV at large k-inplane. Analysis of specific spectra of bare WSe₂ and PL at 260 K might help to understand these two modes (exciton or trions, or polarization-dependence here?). However, based on the direct observation of two parallel modes, it is hardly to draw the conclusion of strong coupling regime at this temperature.

*Our response: We thank the referee for his comment. In the revised version of the manuscript, fig 2c/d depicts the reflectivity measurements of our device to support the strong coupling regime. We have modified the mainbody text accordingly. We have also examined the possibility of having two Tamm-plasmon modes in our structure having parallel dispersions, as suggested by the referee. Basing on the theory developed by M.A. Kaliteevski et al, PRB **76**, 165415 (2010), appearance of two*

modes split by about 20 meV would be possible in the case of the variation of the thickness of the last dielectric layer of the structure by 7-8 nm. This would be very unlikely in our structure. Having no any indication of the existence of the second photonic mode, we rule out this scenario.

In summary, this metal-DBR microcavity structure indeed brings up some alternative platform for the polariton physics in the 2D materials. But the points above need to be thoroughly addressed to support their major claim.

Our response: We thank the referee for pointing out the importance of our work, and are convinced that we could address all raised points in detail.

List of changes:

Fig 2 was modified, in that we have replaced the PL data recorded at 260 K by the room temperature reflectivity data in the strong coupling regime. The fits of the reflectivity data have been refined. The discussion of the figure has been revised accordingly.

Reviewers' comments:

Reviewer #2 (Remarks to the Author):

I appreciate that the authors tried to address the points I raised last time, since I think it is an interesting claim to show the strong coupling regime with this Tamm structure.

But the current manuscript is still not supporting their major claim with one critical point that was quite blurry and was avoided: the real exciton energy of WSe₂, as well as the linewidth at room temperature.

1) Based on the data, Fig. 2b the exciton energy at room temperature is around ~ 1.654 eV, corresponding to PL at Fig. S6, while the reflectivity data take exciton energy around ~ 1.647 eV. The response from the authors regards this as the Stokes shift. The Stokes shift is the absorption energy (reflectivity in this case) minus the PL peak energy. Here they clearly used a smaller energy for the reflectivity, does this correspond to an anti-Stokes shift? So far as I know, the anti-Stokes shift is not reported in WSe₂ (only some reports in the TMD monolayer of MoTe₂ in some 1T phase). The interpretation of the reflectivity data is very misleading, and the splitting fitted with this exciton energy is not convincing.

2) 2D TMD are an emerging group of semiconductors with many interesting opportunities as well as many uncertainties due to the sensitive physical properties. The linewidth of WSe₂ is one of the most sensitive parameters, which is crucial to determine the strong coupling regime when the Rabi splitting is comparable with the linewidth. This manuscript lacks the examination of the linewidth with considering the homogeneous and inhomogeneous broadening as well as the trionic effect. Maybe some temperature-dependent spectral analysis would help.

The 260 K data is very problematic as well, as this figure was totally removed from the manuscript. The $\sim 23\text{meV}$ splitting corresponds to a $\sim 10\text{ nm}$ spectral separation. A mode separation of 10 nm in Fabry-Perot cavity just asks for 3 nm thickness variation if this is a half-lambda cavity with layer refractive index of ~ 1.7 . This variation could be very possible from the monolayer region to few-layer thick region in this cavity. This situation cannot be easily ruled out, and thus may create some uncertainties in the observation of the double modes at this temperature (or even at room temperature).

The major claim based on current data is not sound, and more sufficient evidences (more data or more optimized cavity devices) are needed to support it.

In the following, we reply to all questions/concerns raised by the referee.

I appreciate that the authors tried to address the points I raised last time, since I think it is an interesting claim to show the strong coupling regime with this Tamm structure.

But the current manuscript is still not supporting their major claim with one critical point that was quite blurry and was avoided: the real exciton energy of WSe₂, as well as the linewidth at room temperature.

1) Based on the data, Fig. 2b the exciton energy at room temperature is around $\sim 1.654\text{ eV}$, corresponding to PL at Fig. S6, while the reflectivity data take exciton energy around $\sim 1.647\text{ eV}$. The response from the authors regards this as the Stokes shift. The Stokes shift is the absorption energy (reflectivity in this case) minus the PL peak energy. Here they clearly used a smaller energy for the reflectivity, does this correspond to an anti-Stokes shift? So far as I know, the anti-Stokes shift is not reported in WSe₂ (only some reports in the TMD monolayer of MoTe₂ in some 1T phase). The interpretation of the reflectivity data is very misleading, and the splitting fitted with this exciton energy is not convincing.

- We thank the referee for carefully reviewing our data. Here, we address and clarify the terminology used in the resubmitted version. By 'Stokes shift' we were generically referring to a frequency shift between emission and reflection spectra, irrespectively of the microscopic origin of the effect. We would like to underline again at this point that the reflection data were recorded three month after the PL data, and that the monolayer is capped by a layer of PMMA. Thus, a slight change in the polymer over time (approx. 3 month), for instance swelling due to some moisture in the environment or acetone vapor (solvent used for the heat conducting silver past for the cryo mounting), can easily induce a strain related energy shift. In the literature, gauge factors amounting $-50\text{ meV}/\%$ in WSe₂ have been reported (Schmidt et al. 2D materials 3 (2016), which illustrate that small mechanical changes in the sample can easily lead to energy shifts on the meV scale, as we observe in our experiment. .

We have added a paragraph in supplementary S7 discussing the possible origins of the observed frequency shifts, and a reference to the supplementary informations in the mainbody text.

2) 2D TMD are an emerging group of semiconductors with many interesting opportunities as well as many uncertainties due to the sensitive physical properties. The linewidth of WSe₂ is one of the most sensitive parameters, which is crucial to determine the strong coupling regime when the Rabi splitting is comparable with the linewidth. This manuscript lacks the examination of the linewidth with considering the homogeneous and inhomogeneous broadening as well as the trionic effect. Maybe some temperature-dependent spectral analysis would help.

- In the initial submission, we have already stated that the linewidth of the PL at 300 K is 37.5 meV, and the linewidth of the cavity resonance is 15 meV. In fact, as explained in S4 the PMMA capping can further decrease the excitonic linewidth by a few meV. Therefore, the actual exciton linewidth of the monolayer integrated into the measure structure may even be lightly smaller than 37.5 meV. This is fully consistent with the linewidths of the lower polariton resonance at $k = 0$ of 29.8 meV which ranges between the cavity and the exciton resonance.

As we have exhaustively discussed in the supplementary section of the revised manuscript, the acquired values are in full quantitative agreement with coupled oscillator calculations, taking into account our experimentally extracted parameters.

The 260 K data is very problematic as well, as this figure was totally removed from the manuscript. The ~23meV splitting corresponds to a ~10 nm spectral separation. A mode separation of 10 nm in Fabry-Perot cavity just asks for 3 nm thickness variation if this is a half-lambda cavity with layer refractive index of ~1.7. This variation could be very possible from the monolayer region to few-layer thick region in this cavity. This situation cannot be easily ruled out, and thus may create some uncertainties in the observation of the double modes at this temperature (or even at room temperature).

- As we have explained in the last response letter, we have carried out full transfer matrix calculations in the coupled oscillator framework and reproduced the data with a good accuracy. Since we lack sufficient structural information to confirm or exclude the possibility of two split modes, we have decided to remove the dataset from the mainbody.

The major claim based on current data is not sound, and more sufficient evidences (more data or more optimized cavity devices) are needed to support it.

- We have observed the full polariton dispersion of the lower and upper branches, both in reflection and PL spectra at 300K. This is a fact, which the referee does not challenge. There is only one possible interpretation of such a behavior, which is the strong coupling regime. Thus, we do not understand what brings the referee to the conclusion that the data is not sound.